# Toward a global calibration for quantifying past oxygenation in oxygen minimum zones using benthic Foraminifera

Martin Tetard[1], Laetitia Licari[1], Ekaterina Ovsepyan[2], Kazuyo Tachikawa[1], and Luc Beaufort[1]

[1]Aix Marseille Univ, CNRS, IRD, Coll France, INRAE, CEREGE, Aix-en-Provence, France.
[2]Shirshov Institute of Oceanology, Russian Academy of Sciences, Moscow, Russia.

**Correspondence:** M. Tetard (tetard.martin@gmail.com)

**Abstract.**

Oxygen Minimum Zones (OMZs) are oceanic areas largely depleted in dissolved oxygen, nowadays considered in expansion in the face of global warming. To investigate the relationship between OMZ expansion and global climate changes during the late Quaternary, quantitative oxygen reconstructions are needed, but are still in their early development.

Here, past bottom water oxygenation (BWO) was quantitatively assessed through a new, fast, semi-automated, and taxon-independent morphometric analysis of benthic foraminiferal tests, developed and calibrated using WNP (Western North Pacific, including its marginal seas), ENP (Eastern North Pacific) and the ESP (Eastern South Pacific) OMZs samples. This new approach is based on an average size and shape index for each sample. This method, as well as two already published micropalaeontological techniques based on benthic foraminiferal assemblages variability and porosity investigation of a single

species, were here calibrated based on availability of new data from 45 core tops recovered along an oxygen gradient (from 0.03 to 2.88 mL.L$^{-1}$) from the WNP, ENP, EEP (Eastern Equatorial Pacific), ESP, SWACM (South West African Continental Margin), AS (Arabian Sea) OMZs. Global calibrated transfer functions are thus herein proposed for these methods.

These micropalaeontological reconstruction approaches were then applied on a paleorecord from the ENP OMZ to examine the consistency and limits of these methods, as well as the relative influence of bottom and pore waters on these micropalaeon-

tological tools. Both the assemblages and morphometric approaches (that is also ultimately based on the ecological response of the complete assemblage and faunal succession according to BWO) gave similar and consistent past BWO reconstructions, while the porosity approach (based on a single species and its unique response to a mixed signal of bottom and pore waters) shown ambiguous estimations.

## 1 Introduction

Oxygen Minimum Zones (OMZs) are defined by a dissolved oxygen content of water lower than 0.5 mL.L$^{-1}$ (= 20 $\mu$mol.kg$^{-1}$; Gilly et al., 2013) mainly due to combination of two factors: (1) a high oxygen consumption rate by biotic remineralisation

of the organic matter originating from primary producers during its slow sedimentation in nutrient-rich environments; and (2) a limited physical regeneration due to a slow oceanic circulation (Wyrtki, 1962; Paulmier and Ruiz-Pino, 2009; Gilly et al., 2013; Praetorius et al., 2015). These areas have expanded over the past 50 years (now represent about 10 % of the global oceans' volume), and a further deoxygenation is expected from now until the end of the century (Stramma et al., 2008, 2010;

Gilly et al., 2013; Bopp et al., 2017; Breitburg et al., 2018) as a result of global warming (e.g., Moss et al., 2008). Due to their implication in structuring modern ecosystems, biodiversity, fisheries and their relationship with climate change, carbon pump and carbonate system, the investigation and quantification of past bottom water oxygenation (BWO) are key to understanding their behaviour in a context of global warming and predict their future evolution (Paulmier and Ruiz-Pino, 2009; Gilly et al., 2013; Gilbert, 2017; Breitburg et al., 2018; Levin, 2018).

Benthic foraminifera are considered to be sensitive tracers of temporal and spatial variations of OMZs' intensity (Bernhard and Reimers, 1991; Cannariato and Kennett, 1999) as oxygen is usually the main limiting factor in these areas (Jorissen et al., 1995). Whatever their investigation methods were, most of the studies concerned usually produced qualitative past OMZ reconstructions and did not use the species specific adaptation of benthic foraminifera with regards to oxygen concentration to produce quantitative reconstructions (e.g., den Dulk et al., 1998, 2000; Cannariato and Kennett, 1999; Cannariato et al., 1999;

Ohkushi et al., 2013; Moffitt et al., 2014, 2015a, b). Regarding the benthic foraminiferal species adaptation to dissolved oxygen values usually found in OMZs, the scale historically used to define assemblages corresponds to oxic ($>1.5$ mL.L$^{-1}$), suboxic or intermediate hypoxic (0.5 to 1.4 or 1.5 mL.L$^{-1}$) and dysoxic or severe hypoxic ($< 0.5$ mL.L$^{-1}$) (Kaiho, 1994; Cannariato and Kennett, 1999; Cannariato et al., 1999; Jorissen et al., 2007; Ohkushi et al., 2013; Palmer et al., 2020), and we decided to follow the terminology recently used by Palmer et al. (2020). Our previous investigation focused on two methods based on

assemblages composition (Tetard et al., 2017a) and porosity measurements (Tetard et al., 2017b) to quantitatively reconstruct variations in past BWO in the largest worldwide OMZ, the Eastern North Pacific (ENP) OMZ. However, both methods still required substantial taxonomical knowledge, and are time-consuming. A modern calibration is also still needed.

Several studies already reported morphological (size and shape of tests) responses of benthic foraminifera according to environmental gradients such as oxygenation (Corliss, 1991; Kaiho, 1994; Kaiho et al., 2006). Indeed, the principal idea behind

the use of morphometry for paleo-environmental reconstructions is that benthic foraminiferal shell morphology usually depends on the micro-habitat preferences of each species (Corliss, 1991). Palmer et al. (2020) reminds that in poorly oxygenated environment, benthic foraminiferal faunas are usually dominated by infaunal and elongate species with high porosity while porcelaneous and epifaunal taxa are more abundant in well oxygenated conditions (Kaiho, 1994; Jorissen et al., 1995, 2007). Indeed, on the one hand, epibenthic species (surface dwellers living in oxic waters) are usually more circular (meaning tro-

chospiral and planispiral tests here). Good examples are *Cibicides* and *Planulina* species that are rounded and usually lay flat on the sediment or attached to a substrate in oxygenated conditions. On the other hand, endobenthic species such as *Bolivina* and *Buliminella* thrive under oxygen-depleted conditions and tend to display more elongated (serial) tests (shallow to deep infaunal), allowing them to bury themselves several centimetre-deep into the sediment. When oxic conditions prevail, the benthic fauna is dominated by epibenthic species. A decrease in bottom water oxygenation is usually associated with a replacement of

the epibenthic fauna by deep and shallow endobenthic species moved up to the water / sediment interface following the redox

front (Jorissen et al., 1995, 2007). Such changes were reported by the serial/spiral forms ratio in Core MD02-2508 from the ENP OMZ (Tetard et al., 2017a). An increase in the average roundness of the assemblage is thus likely to indicate more oxic conditions while bottom water conditions progressively depleted in oxygen will show a decrease in roundness (Tetard et al., 2017a). The roundness factor is thus prone to inter-specific, but also to intra-specific (lengthening or widening of individual

5    species in their respective oxygen range, due to an actual widening of chambers or due to a shortening of shells) influences. The overall morphometry of assemblages in OMZs will be further discussed in subsection 4.1.

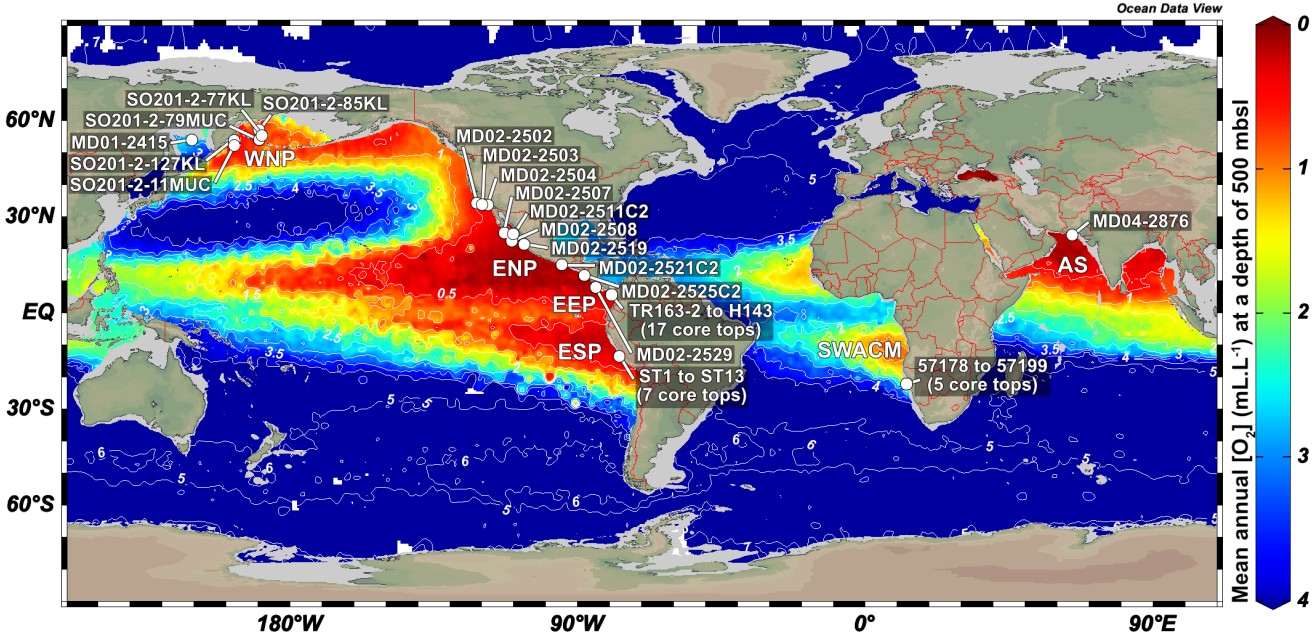

**Figure 1.** Location (white dot) of the different cores used in this study. Labels show core names. Depths and mean annual dissolved oxygen concentrations near core sites are available in Table 1. This figure was generated by using the Ocean Data View software (Schlitzer, Reiner, Ocean Data View, odv.awi.de, 2020) and the World Ocean Atlas 2013 data set (Garcia et al., 2014).

These observations led to the development of a new morphometric method, accessible and easy-to-perform for non-specialists, requiring little equipment, and which is fast and relies on semi-automated size and shape measurements. However, relationships between benthic foraminifera (assemblages, measurements) and environmental parameters are usually region-based and

10    difficult to be applied globally (Palmer et al., 2020). To this aim, 45 core tops recovered worldwide along oxygen gradients from several oxygen deficient areas such as the WNP (Western North Pacific, including its marginal seas), the ENP, the EEP (Eastern Equatorial Pacific), the ESP (Eastern South Pacific), the SWACM (Southwest African Continental Margin), and the AS (Arabian Sea) OMZs, of which the modern values are known, were used for the global calibration of each of the three existing BWO estimation methods, based on either assemblage composition, porosity, or morphometry..

## 2  Material and methods

### 2.1  Core materials

Core MD02-2508, the main core investigated in this study was retrieved $\sim$ 90 km off Baja California coast (latitude: 23°27.91'N; longitude: 111°35.74'W) during the R/V *Marion-Dufresne* MD126 MONA (IMAGES VII) campaign in 2002 (Fig.1a). This
40.42 m core was piston-cored (giant corer Calypso) at a depth of 606 mbsl and covers the last 80 kyr (Blanchet et al., 2007). At this location and depth, the coring site lays in margin of the ENP OMZ core, where the current mean annual $[O_2]$ is about 0.13 mL.L$^{-1}$ according to the World Ocean Atlas 2013 dataset (Garcia et al., 2014, http://www.nodc.noaa.gov/OC5/woa13/).

Several core tops, recovered along transects through the WNP, ENP, EEP, ESP, SWACM, and AS OMZs using piston-corer, multi-corer and CASQ-corer were also investigated. This allowed the sampling throughout an oxygenation gradient of 0.03
to 2.88 mL.L$^{-1}$, where bottom water concentrations were estimated using the World Ocean Atlas 2013 (Garcia et al., 2014) regarding the ENP, AS and WNP core sites, the World Ocean Atlas 2009 (Garcia et al., 2010) regarding the EEP core sites (originally investigated by Bandy and Arnal (1957); Betancur and Martinez (2003) and recently by Patarroyo and Martinez (2021)), and CTDO measurements concerning the ESP and SWACM (investigated by Leiter and Altenbach (2010)) core sites. Core tops location, water depth, and modern $[O_2]$ values are detailed in Table 1. Only 1-centimetre sections that usually
have less than a few hundreds of years were investigated. These cores are thus suitable for faunal comparison and calibration purposes.

The ENP core tops (See Table 1) were carried out at CEREGE (preparation, census counting of benthic foraminifera and morphometric measurements), except for cores MD02-2503 and MD02-2504 (Ohkushi et al., 2013) and Core MD02-2529 (Ovsepyan and Ivanova., 2009). The ESP Peruvian margin OMZ core tops (ST1 to ST13) were recovered during the April
2015 CRIO campaign on the R/V *Olaya* where sediment sampling and CTDO measurements were carried out on five sites along the Callao transect (ST1 to ST5), and two sites along the Pisco transect (ST12 to ST13) and also prepared and processed at CEREGE. All the investigation on the core tops recovered from the WNP were carried out by Ekaterina Ovsepyan (regarding SO201-2-11MUC, SO201-2-77KL, SO201-2-79MUC, SO201-2-127KL) and Bubenshchikova et al. (2015, regarding MD01-2415, dataset of benthic foraminiferal counts is available in their supplementary material). These samples were collected
during the R/V *Sonne* cruise Leg SO201-2 in 2009 and the WEPAMA 2001 cruise of the R/V *Marion Dufresne*, respectively. Regarding the EEP OMZ, census counting and oxygen information from 17 core tops (see list in Table 1), recovered from the Panama Basin, originally investigated by Bandy and Arnal (1957); Betancur and Martinez (2003) were recovered from the supplementary material shared by Patarroyo and Martinez (2021). Regarding the SWACM samples, as no census data are provided in Leiter and Altenbach (2010), we used the relative abundance visible on their Figs 3 and 5. As the authors divided
their samples into a 150-250 and a >250 $\mu$m size fractions, and provided only benthic foraminiferal relative abundances for each fraction of each sample without giving the total number of tests per sample, we were not able to gather both fractions to investigated the >150 $\mu$m fraction. Thus, only 5 samples were investigated, where the relative abundance for the 150-250 $\mu$m fraction (their Fig. 5) was available and where no foraminifera were present in the >250 $\mu$m fraction (their Fig. 3). Finally, Core MD04-2876 recovered from the AS OMZ was investigated by Laetitia Licari.

**Table 1.** Core station location, depth and modern dissolved oxygen level for each investigated core top sample.

| Core top station | OMZ | Site location (latitude) | Site location (longitude) | Water depth (mbsl) | [O$_2$] measurement method | Modern [O$_2$] mL.L$^{-1}$ | Investigated size fraction ($\mu$m) | Author (name, year) |
|---|---|---|---|---|---|---|---|---|
| MD01-2415 | WNP | 53$^{\circ}$57.09'N | 149$^{\circ}$57.52'E | 822 | WOA2013 | 1.03 | >125 | Bubenshchikova et al. (2015) |
| SO201-2-11MUC | WNP | 53$^{\circ}$59.47'N | 162$^{\circ}$22.53'E | 2169 | WOA2013 | 1.79 | >63 | This study |
| SO201-2-77KL | WNP | 56$^{\circ}$19.90'N | 170$^{\circ}$41.97'E | 2163 | WOA2013 | 1.65 | >63 | Ovsepyan et al. (2021) |
| SO201-2-79MUC | WNP | 56$^{\circ}$42.99'N | 170$^{\circ}$29.78'E | 1161 | WOA2013 | 0.83 | >63 | This study |
| SO201-2-127KL | WNP | 54$^{\circ}$23.66'N | 162$^{\circ}$13.34'E | 1440 | WOA2013 | 1.14 | >63 | This study |
| MD02-2502 | ENP | 34$^{\circ}$39.94'N | 120$^{\circ}$58.03'W | 404 | WOA2013 | 0.83 | >150 | This study |
| MD02-2503 | ENP | 34$^{\circ}$17.17'N | 120$^{\circ}$02.19'W | 569 | WOA2013 | 0.33 | >63 | Ohkushi et al. (2013) |
| MD02-2504 | ENP | 34$^{\circ}$13.99'N | 119$^{\circ}$52.08W | 481 | WOA2013 | 0.44 | >63 | Ohkushi et al. (2013) |
| MD02-2507 | ENP | 25$^{\circ}$08.00'N | 112$^{\circ}$42.09'W | 495 | WOA2013 | 0.25 | >150 | This study |
| MD02-2508 | ENP | 23$^{\circ}$27.91'N | 111$^{\circ}$35.74'W | 606 | WOA2013 | 0.13 | >150 | This study |
| MD02-2511C2 | ENP | 24$^{\circ}$38.74'N | 110$^{\circ}$36.12'W | 417 | WOA2013 | 0.16 | >150 | This study |
| MD02-2519 | ENP | 22$^{\circ}$30.89'N | 106$^{\circ}$39.00'W | 955 | WOA2013 | 0.21 | >150 | This study |
| MD02-2521C2 | ENP | 15$^{\circ}$40.25'N | 95$^{\circ}$18.00'W | 718 | WOA2013 | 0.09 | >150 | This study |
| MD02-2525C2 | ENP | 12$^{\circ}$00.47'N | 87$^{\circ}$54.44'W | 877 | WOA2013 | 0.28 | >150 | This study |
| MD02-2529 | ENP | 8$^{\circ}$12.33'N | 84$^{\circ}$07.32'W | 1661 | WOA2013 | 1.65 | >150 | Ovsepyan and Ivanova. (2009) |
| TR 163-2 | EEP | 8$^{\circ}$15.00'N | 84$^{\circ}$18.00'W | 1620 | WOA2009 | 1.93 | >150 | Betancur and Martinez (2003) |
| TR 163-11 | EEP | 6$^{\circ}$27.00'N | 85$^{\circ}$48.00'W | 1950 | WOA2009 | 2.25 | >150 | Betancur and Martinez (2003) |
| TR 163-13 | EEP | 6$^{\circ}$01.80'N | 87$^{\circ}$24.00'W | 2450 | WOA2009 | 2.52 | >150 | Betancur and Martinez (2003) |
| TR 163-15 | EEP | 6$^{\circ}$16.20'N | 87$^{\circ}$54.00'W | 1770 | WOA2009 | 2.07 | >150 | Betancur and Martinez (2003) |
| TR 163-26 | EEP | 1$^{\circ}$53.40'N | 87$^{\circ}$48.00'W | 3000 | WOA2009 | 2.88 | >150 | Betancur and Martinez (2003) |
| TR 163-33 | EEP | 1$^{\circ}$54.60'N | 82$^{\circ}$36.00'W | 2230 | WOA2009 | 2.33 | >150 | Betancur and Martinez (2003) |
| TR 163-34 | EEP | 1$^{\circ}$18.60'N | 81$^{\circ}$54.00'W | 1360 | WOA2009 | 1.80 | >150 | Betancur and Martinez (2003) |
| TR 163-35 | EEP | 1$^{\circ}$21.00'N | 81$^{\circ}$54.00'W | 1415 | WOA2009 | 1.89 | >150 | Betancur and Martinez (2003) |
| TR 163-36 | EEP | 1$^{\circ}$21.36'N | 81$^{\circ}$48.00'W | 1780 | WOA2009 | 2.24 | >150 | Betancur and Martinez (2003) |
| TR 163-37 | EEP | 1$^{\circ}$21.00'N | 81$^{\circ}$42.00'W | 2005 | WOA2009 | 2.35 | >150 | Betancur and Martinez (2003) |
| TR 163-38 | EEP | 1$^{\circ}$20.40'N | 81$^{\circ}$36.00'W | 2200 | WOA2009 | 2.56 | >150 | Betancur and Martinez (2003) |
| ODP 84 | EEP | 5$^{\circ}$45.00'N | 82$^{\circ}$54.00'W | 3096 | WOA2009 | 2.49 | >150 | Betancur and Martinez (2003) |
| ODP 506B | EEP | 0$^{\circ}$36.60'N | 86$^{\circ}$06.00'W | 2711 | WOA2009 | 2.47 | >150 | Betancur and Martinez (2003) |
| H37 | EEP | 7$^{\circ}$06.00'N | 78$^{\circ}$18.00'W | 1400 | WOA2009 | 1.44 | >61 | Bandy and Arnal (1957) |
| H130 | EEP | 9$^{\circ}$09.00'N | 84$^{\circ}$09.00'W | 1246 | WOA2009 | 1.20 | >61 | Bandy and Arnal (1957) |
| H141 | EEP | 6$^{\circ}$27.00'N | 81$^{\circ}$00.00'W | 1912 | WOA2009 | 1.85 | >61 | Bandy and Arnal (1957) |
| H143 | EEP | 7$^{\circ}$06.00'N | 80$^{\circ}$28.20'W | 1025 | WOA2009 | 1.00 | >61 | Bandy and Arnal (1957) |
| ST1 | ESP | 12$^{\circ}$01.77'S | 77$^{\circ}$12.86'W | 44 | CTDO | 0.03 | >150 | This study |
| ST2 | ESP | 12$^{\circ}$02.48'S | 77$^{\circ}$17.29'W | 90 | CTDO | 0.04 | >150 | This study |
| ST3 | ESP | 12$^{\circ}$02.41'S | 77$^{\circ}$22.58'W | 110 | CTDO | 0.03 | >150 | This study |
| ST4 | ESP | 12$^{\circ}$02.92'S | 77$^{\circ}$29.12'W | 140 | CTDO | 0.09 | >150 | This study |
| ST5 | ESP | 12$^{\circ}$02.60'S | 77$^{\circ}$39.34'W | 165 | CTDO | 0.08 | >150 | This study |
| ST12 | ESP | 14$^{\circ}$05.04'S | 76$^{\circ}$26.79'W | 175 | CTDO | 0.04 | >150 | This study |
| ST13 | ESP | 14$^{\circ}$02.46'S | 76$^{\circ}$22.02'W | 67 | CTDO | 0.05 | >150 | This study |
| 57178 | SWACM | 23$^{\circ}$45.51'S | 14$^{\circ}$15.99'E | 114 | CTDO | 0.40 | >150 | Leiter and Altenbach (2010) |
| 57184 | SWACM | 23$^{\circ}$00.02'S | 14$^{\circ}$22.05'W | 44 | CTDO | 0.50 | >150 | Leiter and Altenbach (2010) |
| 57187 | SWACM | 23$^{\circ}$00.05'S | 14$^{\circ}$02.92'W | 130 | CTDO | 0.20 | >150 | Leiter and Altenbach (2010) |
| 57198 | SWACM | 21$^{\circ}$45.79'S | 13$^{\circ}$43.41'W | 93 | CTDO | 0.50 | >150 | Leiter and Altenbach (2010) |
| 57199 | SWACM | 22$^{\circ}$00.22'S | 13$^{\circ}$51.50'W | 91 | CTDO | 0.30 | >150 | Leiter and Altenbach (2010) |
| MD04-2876 | AS | 24$^{\circ}$50'57N | 064$^{\circ}$00'49E | 828 | WOA2013 | 0.17 | >150 | This study |

## 2.2 Benthic foraminiferal assemblages study

Regarding the preparation of the 14 core top samples available at CEREGE for calibration purposes (seven from the ENP (MD02-2502, MD02-2507, MD02-2508, MD02-2511C2, MD02-2519, MD02-2521C2, MD02-2525C2) and seven from the ESP (ST1, ST2, ST3, ST4, ST5, ST12, ST13) transects), these samples usually contained benthic foraminifera often encrusted in organic matter-rich clays. To extract benthic foraminiferal tests (as OMZ samples are mostly composed of calcareous tests, and very few agglutinated or soft-shelled specimens which are usually not preserved in the fossil record; Gooday et al., 2000)), the following procedure was adapted from Bairbakish et al. (1999):

   (1) A 1 cm-thick slice was sampled and weighed.

(2) The sample was oven-dried during 48 hours and weighed again as a routine laboratory procedure to get estimate of wet and dry weights of bulk sediment.

(3) The sample was gently re-wet to disaggregate sediment mass using a water-spray over 63 and 150 $\mu$m sized meshes without damaging the foraminiferal tests.

(4) If the sediments did not disaggregate, >63 $\mu$m residues were placed into a 25 mL beaker where 6 mL of $\sim$ 3 % diluted NaClO (Sodium Hypochlorite solution) and 3 mL of $\sim$ 35 % diluted $H_2O_2$ (Hydrogen Peroxide solution) were added during 10 min to disaggregate organic clots. Residues were then rinsed with tap water and sieved over 63 and 150 $\mu$m sized meshes.

(5) The content of each sieve was rinsed with distilled water and filtered before being oven-dried and stored in vials.

(6) Fine (63-150 $\mu$m) and coarse (>150 $\mu$m) fractions were weighed. Only the coarse fraction was investigated herein.

The 31 other core tops used for the calibration of the assemblages methods were previously counted for benthic foraminifera by E. Ovsepyan (MD02-2529, SO201-2-11MUC, SO201-2-77KL, SO201-2-79MUC, SO201-2-127KL), Bubenshchikova et al. (2015, MD01-2415), Ohkushi et al. (2013, MD02-2503 and MD02-2504), Betancur and Martinez (2003); Patarroyo and Martinez (2021, TR 163-2, TR 163-11, TR 163-13, TR 163-15, TR 163-26, TR 163-33, TR 163-34, TR 163-35, TR 163-36, TR 163-37, TR 163-38, ODP 84, ODP 506B), Bandy and Arnal (1957); Patarroyo and Martinez (2021, H37, H130, H141, H143), Leiter and Altenbach (2010, 57178, 57184, 57187, 57198, 57199) and L. Licari (MD04-2876). The benthic foraminiferal specimens used in this study for Core MD02-2508 were already picked for assemblage investigation in Tetard et al. (2017a). Depending on the abundance of benthic foraminifera, different aliquots were investigated to pick about 300 specimens per sample. We decided to mainly focus on coarse fraction data (>150 $\mu$m) as it is the size fraction commonly used in benthic foraminiferal studies and in order to propose a calibration adapted to most of the studies. However, we also decided to include fine fraction data in our calibration by using the >63 $\mu$m counting when available (for 10 out of the 45 core tops; see Table 2). According to this, only 17 out of the 63 shared core tops census counting from Patarroyo and Martinez (2021) were used (the ones originally investigated by Bandy and Arnal (1957) and Betancur and Martinez (2003)), as the other core tops were investigated for the fine fraction alone.

## 2.3   Pore density analysis

Regarding the Pacific basin, *Bolivina seminuda* from the ENP was previously investigated for its porosity (Tetard et al., 2017b). This species was very abundant throughout Core MD02-2508, and is also present in two other ENP core tops (MD02-2519 and MD02-2525C2) and its ecological adaptation is well documented in the literature (e.g. Glock et al., 2013). As a consequence, it was selected in order to associate porosity values to oxygenation. The original procedure consists in picking 30 specimens which are then gently crushed between two microscope glass slides. The test fragments are then dropped along with ethanol into a compartmented decanter for a uniform and random settling of particles on a microscope glass slide. After drying, cover glasses are placed along with optical adhesive on each sample of the glass slide. Using a customised image acquisition software, images of each fragments are automatically acquired under a transmitted light microscope and send to an image processing software that binarises it and computes the area of each fragment and its pores. The average porosity for each sample could then be investigated (see detailed procedure in Tetard et al. (2017b)).

## 2.4 Morphometric analysis procedure

As no tests were altered or broken (only some moderate in-situ dissolution was observed in a few samples; absolute preservation score of 6 to 8 on Nguyen et al. (2009)'s preservation scale), we were able to observe and measure several size and shape indices. Indeed, as the environmental conditions in OMZ usually correspond to oxygen-depleted clays with very limited bioturbation, shells are most of the time very well preserved. Moreover, we believe that good preservation might also be related to high sedimentation rates at the site locations studied, most likely due to their close position to the land where they could be affected by enhanced terrigenous material supply. Although the delicate shell of typical OMZ species such as *Bolivina* and *Eubulimina* species might appear fragile and delicate, these tests are quite tough and we are confident that our cleaning procedure did not affect their preservation, assemblage composition overall, and morphology of the shells. The taxon-independent and semi-automated morphometric analysis was performed on 16 samples (three from the WNP (SO201-2-11MUC, SO201-2-79MUC, SO201-2-127KL), six from the ENP (MD02-2502, MD02-2508, MD02-2511C2, MD02-2519, MD02-2521C2, MD02-2525C2), six from the ESP (ST1, ST2, ST3, ST4, ST5, ST12), and one from the AS (MD04-2876) transects). As the fine fraction (63 - 150 $\mu$m) might exhibits fragmented and broken tests (which do not reflect the true morphology of the organisms when they were living) or juvenile specimens (which do not always reflect the true morphology of the adult forms (e.g. for macrospheric forms of *Bolivina* species, the proloculus is usually very rounded while the adult specimen can be very elongated)), this fraction was not investigated for the morphometric analysis which was focused on the >150 $\mu$m fraction.

In order to measure several morphometric parameters for the complete assemblage of each sample, every specimen picked in Tetard et al. (2017a) study (around 300 specimens per sample, corresponding to the number of specimens needed for representing a complete assemblage; Buzas (1990); Fatela and Taborda (2002)) was dropped on a 20 × 20 mm black square and a single picture per sample was acquired under a stereoscopic microscope. The light should be bright enough for the tests to be well visible on the black background, but not too bright so the background stays completely dark (see Fig. 2a as an example). These settings are the best compromise between resolution and spacing between every test for preventing contact.

The following procedure was used for the automated morphometric processing of every sample image on the ImageJ image analysis software (v.1.52e Schneider et al., 2012, http://imagej.nih.gov/ij/), and automatically performed by using the MorFo_.ijm (**Mor**phometrics on **Fo**raminifera) java macro developed for this study (available at: https://github.com/microfossil/ImageJ-LabView-Scripts and in supplement). Once installed (simply put the MorFo_.ijm file into the plugin directory of your ImageJ software) and selected, the MorFo plugin will automatically:

(1) Ask the user for the input folder where original images are located and create three output subfolders for resized images, processed images, and results, respectively, into the designated input folder. It also asks the user for a knows distance in pixel and a known distance in $\mu$m in order to generate calculated measurements in $\mu$m.

(2) Open each image individually, reduce its size to 1000 × 1000 pixels, and convert it into an 8-bit image for binarisation in order to obtain a black and white only image (Fig. 2).

(3) Separate the specimens in contact with each other using a "Watershed Irregular Features", performed by using the BioVoxxel plugin (available online at http://fiji.sc/BioVoxxel_Toolbox).

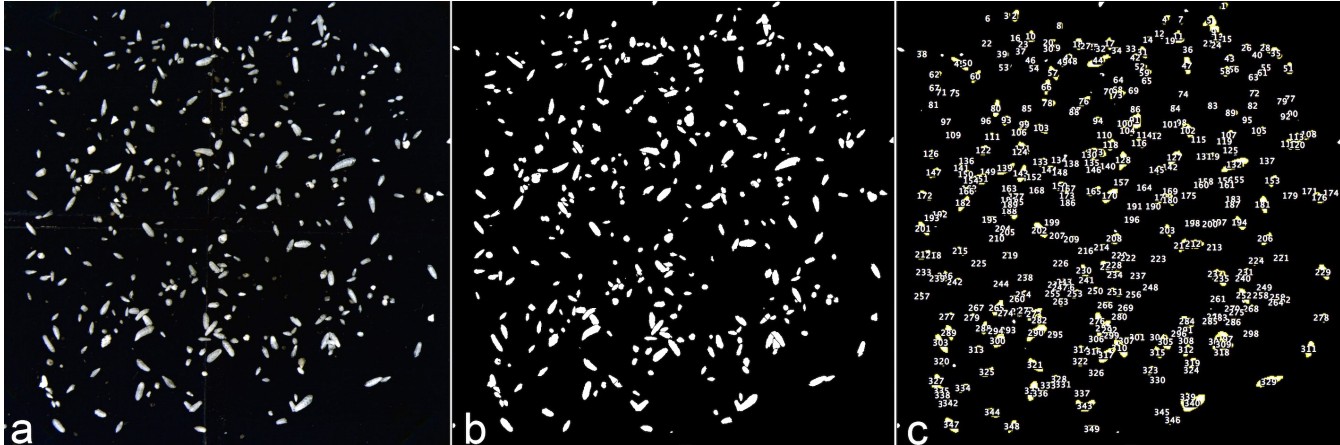

**Figure 2.** Principal steps of the automated image analysis within the *MorFo* macro. a. Original stereoscopic microscope image. b. Binarised image and watershed. c. Automated counting and measurements. The illustrated sample originates from Core MD02-2508, sample 240-241 cm

(4) Select "Area", "Shape descriptors", and "Fit ellipse" within the "Set measurements" panel in order to measure a size and a shape parameter for every specimen.

(5) Run an "Analyze particles" operation ("Size" and "Circularity" are adjusted, here 40-infinity and 0.30-1.00 respectively, so as to exclude small and elongated particles such as dust) to count and measure each specimen.

(6) Save the resized image, processed image, average results for each sample and detail results for each specimen.

(7) Display a success message and automatically close the ImageJ distribution.

The shape descriptor retained in this study is the *Roundness index*, defined as:

$$Roundness\ index = \frac{4 \times Area}{\pi \times Major\ axis^2} \tag{1}$$

This *Roundness index* should then be corrected for size due to the presence of round species occurring in dysoxic conditions (e.g. *Takayanagia delicata* in the WNP ans ENP samples). As these species are relatively small by comparison with large and round epibenthic species characteristic of well oxygenated environments (such as *Cibicides*-like species, or *Hoeglandina elegans*), more or less oxygenated periods can thus be distinguished by taking the size of species into account. The retained size descriptor, here "Major", corresponds to the primary axis of the best fitting ellipse that have the same area as the investigated specimen. This is used to avoid error in measurement in case that some pixels that do not correspond to the specimen (error in the watershed, image artifact) might biased size measurement.

Other size descriptors (e.g. *Feret's diameter* or *Area*) and shape descriptors (e.g. *Circularity* or *Length-to-width ratio*) can also be used (see the *ImageJ User Guide, IJ 1.46r* by Tiago Ferreira and and Wayne Rasband). In order to take simultaneously

both the size and shape descriptors into account into the Pacific OMZs, the morphometric index *MARIN* (**M**ajor **A**xis and **R**oundness **IN**dex) is then calculated:

$$MARIN\ (\mu m) = Major\ axis\ \times\ Roundness \tag{2}$$

This morphometric approach is particularly worthy of consideration for non-specialists as only a basic taxonomic knowledge is required. Benthic foraminifera only need to be picked and imaged under a stereoscopic microscope (a single picture per sample). Images are then automatically processed and no identification is necessary. Approximately 10 min are needed for the processing (picking and imaging) of each sample, while at least 1 hour is usually required for picking and identifying every specimen of each sample with a standard approach.

## 3  Results

### 3.1  Re-calibration of the [O$_2$] estimation by the assemblages method

A method to quantify past oxygenation within the ENP OMZ, based on the relative abundance and succession of three benthic foraminifera assemblages and the oxygenation adaptation of their affiliated species (indicatives of dysoxic, suboxic, and oxic conditions) was developed by Tetard et al. (2017a). This method was originally calibrated using the theoretical [O$_2$] threshold between a 100 % dysoxic assemblage (0.1 mL.L$^{-1}$), 50 % dysoxic and 50 % suboxic (0.5 mL.L$^{-1}$) and 50 % suboxic and 50 % oxic (1.4 mL.L$^{-1}$). However, estimated [O$_2$] values tested on seven Pacific core tops using the assemblages method from Tetard et al. (2017a) show that this original approach seems to be slightly over-estimating oxygenation by comparison with the average oxygen concentration near each site (Garcia et al., 2014).

As a consequence, we decided to use all the available census data from the 45 core tops (five from the WNP, 10 from the ENP, 17 from the EEP, seven from the ESP, five from the SWACM, and one from the AS; see Table 2) of which the modern [O$_2$] values are known (from Garcia et al., 2010, 2014, and CTDO measurements) to propose a global calibration for the assemblages [O$_2$] estimation method based on actual [O$_2$] and not theoretical thresholds. The oxygen measurements available for calibration range from 0.03 to 2.88 mL.L$^{-1}$ (see Table 1) and is compared with a term defined by the Equation 1 and referred to as the benthic foraminiferal assemblage (BFA) index. This equation consists in assigning a number to each sample based on its dysoxic, suboxic, and oxic benthic foraminiferal composition (from 100 if the sample is 100 % composed of dysoxic specimens, to 0 if the sample is 100 % composed of suboxic specimens, to -100 if the sample is composed of 100 % of oxic specimens; see Fig. 3a).

$$Benthic\ foraminiferal\ assemblage\ (BFA)\ index = \%\ dysoxic - \%\ oxic \tag{3}$$

**Table 2.** Benthic foraminiferal assemblage composition, morphometry porosity, and their respective [O$_2$] estimation for each investigated core top sample.

| Core top station | Dysoxic assemblages (%) | Suboxic assemblages (%) | Oxic assemblages (%) | BFA | Estimated [O$_2$] mL.L$^{-1}$ (calibrated assemblages method) | Average size (major axis in $\mu$m) | Average roundness | MARIN | Estimated [O$_2$] mL.L$^{-1}$ (calibrated morphometry method) | PSA (B. seminuda) | Estimated [O$_2$] mL.L$^{-1}$ (calibrated porosity method) |
|---|---|---|---|---|---|---|---|---|---|---|---|
| MD01-2415 | 8.74 | 23.06 | 68.20 | -59.47 | 1.12 | / | / | / | / | / | / |
| SO201-2-11MUC | 1.50 | 1.50 | 97.01 | -95.51 | 2.40 | 424.63 | 0.75 | 319.61 | 2.03 | / | / |
| SO201-2-77KL | 8.19 | 4.09 | 87.72 | -79.53 | 1.71 | / | / | / | / | / | / |
| SO201-2-79MUC | 37.50 | 2.08 | 60.42 | -22.92 | 0.52 | 397.90 | 0.70 | 280.39 | 0.55 | / | / |
| SO201-2-127KL | 8.60 | 19.35 | 72.04 | -63.44 | 1.22 | 413.81 | 0.75 | 311.07 | 1.53 | / | / |
| MD02-2502 | 0.64 | 87.86 | 11.50 | -10.86 | 0.40 | 421.61 | 0.67 | 282.56 | 0.59 | Not enough material | Not enough material |
| MD02-2503 | 0.00 | 99.29 | 0.71 | -0.71 | 0.32 | / | / | / | / | / | / |
| MD02-2504 | 0.00 | 97.73 | 2.27 | -2.27 | 0.33 | / | / | / | / | / | / |
| MD02-2507 | 41.45 | 41.82 | 16.73 | 24.73 | 0.19 | / | / | / | / | Not enough material | Not enough material |
| MD02-2508 | 80.91 | 6.15 | 12.94 | 67.96 | 0.08 | 464.59 | 0.51 | 237.12 | 0.13 | 0.019 | 0.16 |
| MD02-2511C2 | 44.90 | 48.48 | 6.61 | 38.29 | 0.14 | 368.29 | 0.63 | 233.18 | 0.11 | Not enough material | Not enough material |
| MD02-2519 | 24.85 | 53.85 | 21.30 | 3.55 | 0.29 | 366.30 | 0.69 | 253.45 | 0.22 | 0.019 | 0.17 |
| MD02-2521C2 | 56.55 | 9.23 | 34.23 | 22.32 | 0.20 | 356.30 | 0.61 | 217.64 | 0.07 | Not enough material | Not enough material |
| MD02-2525C2 | 29.70 | 46.67 | 23.64 | 6.06 | 0.28 | 352.29 | 0.70 | 248.11 | 0.19 | 0.016 | 0.29 |
| MD02-2529 | 0.74 | 22.06 | 77.21 | -76.47 | 1.60 | / | / | / | / | / | / |
| TR 163-2 | 0.91 | 7.06 | 92.03 | -91.12 | 2.19 | / | / | / | / | / | / |
| TR 163-11 | 1.77 | 1.11 | 97.12 | -95.35 | 2.39 | / | / | / | / | / | / |
| TR 163-13 | 0.00 | 0.00 | 100.00 | -100 | 2.64 | / | / | / | / | / | / |
| TR 163-15 | 0.41 | 0.00 | 99.59 | -99.18 | 2.60 | / | / | / | / | / | / |
| TR 163-26 | 1.21 | 0.00 | 98.79 | -97.59 | 2.51 | / | / | / | / | / | / |
| TR 163-33 | 4.65 | 0.97 | 94.37 | -89.72 | 2.12 | / | / | / | / | / | / |
| TR 163-34 | 0.00 | 3.21 | 96.79 | -96.79 | 2.47 | / | / | / | / | / | / |
| TR 163-35 | 0.40 | 2.22 | 97.38 | -96.97 | 2.48 | / | / | / | / | / | / |
| TR 163-36 | 4.46 | 3.04 | 92.49 | -88.03 | 2.05 | / | / | / | / | / | / |
| TR 163-37 | 3.05 | 2.85 | 94.11 | -91.06 | 2.18 | / | / | / | / | / | / |
| TR 163-38 | 3.71 | 7.20 | 89.09 | -85.39 | 1.94 | / | / | / | / | / | / |
| ODP 84 | 6.13 | 0.00 | 93.87 | -87.74 | 2.04 | / | / | / | / | / | / |
| ODP 506B | 0.71 | 0.00 | 99.29 | -98.57 | 2.56 | / | / | / | / | / | / |
| H37 | 2.40 | 26.80 | 70.70 | -68.30 | 1.35 | / | / | / | / | / | / |
| H130 | 1.50 | 44.10 | 54.40 | -52.20 | 0.97 | / | / | / | / | / | / |
| H141 | 0.00 | 35.50 | 64.50 | -64.50 | 1.24 | / | / | / | / | / | / |
| H143 | 6.50 | 28.60 | 64.90 | -58.40 | 1.09 | / | / | / | / | / | / |
| ST1 | 92.40 | 6.45 | 1.15 | 91.24 | 0.05 | 306.88 | 0.69 | 212.14 | 0.06 | / | / |
| ST2 | 99.03 | 0.97 | 0.00 | 99.03 | 0.04 | 324.14 | 0.69 | 224.15 | 0.08 | / | / |
| ST3 | 94.18 | 5.63 | 0.19 | 94.00 | 0.04 | 307.34 | 0.67 | 204.87 | 0.04 | / | / |
| ST4 | 83.50 | 8.74 | 7.77 | 75.73 | 0.06 | 369.82 | 0.57 | 210.24 | 0.05 | / | / |
| ST5 | 58.70 | 23.77 | 17.53 | 41.17 | 0.13 | 437.17 | 0.56 | 246.31 | 0.18 | / | / |
| ST12 | 75.93 | 2.78 | 21.30 | 54.63 | 0.10 | 361.82 | 0.62 | 224.86 | 0.09 | / | / |
| ST13 | 97.65 | 0.00 | 2.35 | 95.29 | 0.04 | / | / | / | / | / | / |
| 57178 | 13.41 | 86.59 | 0.00 | 13.41 | 0.24 | / | / | / | / | / | / |
| 57184 | 0.00 | 50.09 | 49.91 | -49.91 | 0.91 | / | / | / | / | / | / |
| 57187 | 5.07 | 94.93 | 0.00 | 5.07 | 0.28 | / | / | / | / | / | / |
| 57198 | 0.00 | 100.00 | 0.00 | 0.00 | 0.32 | / | / | / | / | / | / |
| 57199 | 9.24 | 90.76 | 0.00 | 9.24 | 0.26 | / | / | / | / | / | / |
| MD04-2876 | 47.42 | 49.24 | 3.34 | 44.07 | 0.12 | 417.03 | 0.58 | 241.32 | 0.15 | / | / |

The original extrapolation method was refined and now consists in the Equation 2 where the relative abundance of the dysoxic and oxic assemblages (through the use of the benthic foraminiferal assemblage index) can directly be used to determine past oxygenation:

$$[O_2]_{(assemblages\ method)} = 0.317 \times \exp^{-0.0212 \times BFA\ index} \tag{4}$$

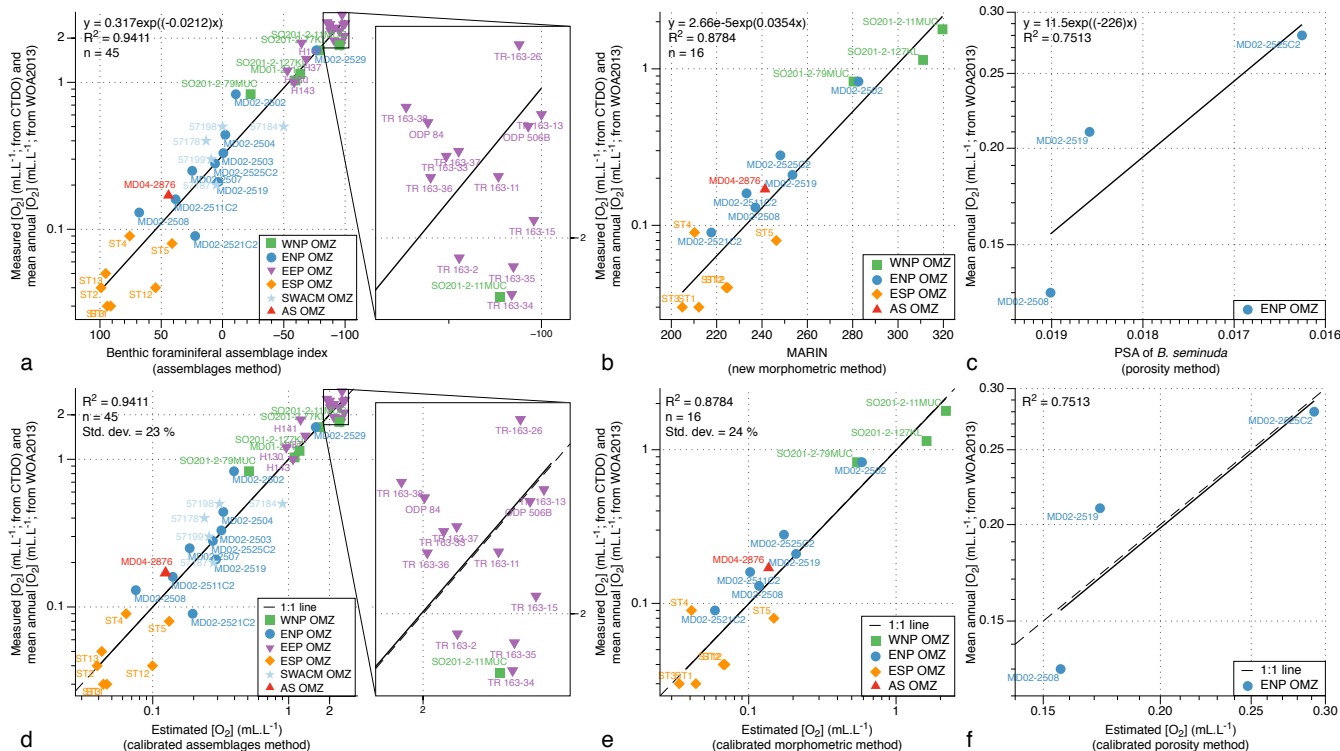

**Figure 3.** a, b, c: Relationship between the bottom water oxygenation and the benthic foraminiferal index, the new morphometric index (*MARIN*) and the porosity of *B. seminuda*. d, e, f: Relationship between the bottom water oxygenation and the calibrated assemblages method estimation, taxon-independent morphometric method estimation, and porosity method estimation.

This equation is based on the consistent relationship ($R^2 = 0.94$, see Fig. 3a) between the benthic foraminiferal assemblage index and extrapolated and measured [$O_2$] values (from Garcia et al., 2010, 2014, and CTDO measurements). Overall, poorly oxygenated samples are associated with high benthic foraminiferal assemblage index values (thus composed of dysoxic species) while well oxygenated core tops are associated with negative benthic foraminiferal assemblage index values (thus showing oxic

5     species). To make these oxygen reconstructions more user-friendly, an excel spreadsheet is provided in supplement (Table S1) to automatically compute past oxygenation by filling at least two of the three columns corresponding to the relative abundance of the dysoxic, suboxic and oxic assemblages. The list of dysoxic and suboxic benthic foraminiferal species used in this study as well as their respective references for assignation to the dysoxic or suboxic assemblage is provided in supplement (Table S2).

10     This approach produces accurate core top samples [$O_2$] estimates that are very close to the modern mean annual [$O_2$] values (Fig. 3d, regression line very close to the 1:1 line; $R^2 = 0.94$) for the WNP, ENP, EEP, ESP, SWACM, and AS OMZs altogether. It is thus likely to be used as a global index in oxygen-deficient areas from different OMZs.

### 3.2 Calibration of the porosity indices

In order to calibrate the procedure described in Tetard et al. (2017b), a microscope slide containing *B. seminuda* fragments was prepared for the three Pacific core tops where this species was present in the first centimetre (MD02-2508, MD02-2519, MD02-2525C2). Considering the limited number of samples, the *PSA* (pore surface area) for these core tops shows a good correlation ($R^2 = 0.75$; Fig. 3c) to the [O$_2$] values near these stations (Garcia et al., 2014). Higher values of BWO is associated with a lower PSA of *B. seminuda* while a higher porosity index is observed together with lower BWO values. The following Equation 2 is then used to extrapolate [O$_2$] for Core MD02-2508 based on its *PSA* values downcore (Fig. 3f):

$$[O_2]_{(porosity\ method)} = 11.5 \times \exp^{-226 \times PSA_{(B.\ seminuda)}} \tag{5}$$

### 3.3 Calibration of the morphometric indices

In order to associate morphometric measurements to BWO concentrations, three core tops available at the Shirshov Institute of Oceanology for the WNP OMZ, eight core tops available at CEREGE from the ENP and AS OMZs, of which the modern annual mean [O$_2$] are known (Garcia et al., 2014), as well as six core tops from the ESP transects (when enough material was available, and not coloured by rose bengal) of which the modern [O$_2$] are known (CTDO measurements) were used for calibration purposes. These samples show lower *MARIN* values during dysoxic conditions and higher values during oxic conditions. The significant correlation ($R^2 = 0.88$) between the estimated [O$_2$] (Garcia et al., 2014) and the corresponding *MARIN* values (Fig. 3b) can then be used to extrapolate past BWO concentrations based on the *MARIN* values for each sample (Fig. 3e), according to the Equation 4:

$$[O_2]_{(morphometric\ method\ for\ the\ ENP\ OMZ)} = 0.0000266 \times \exp^{0.0354 \times MARIN} \tag{6}$$

### 3.4 Estimating the measurement uncertainties of the calibrated methods

In order to estimate the uncertainty linked with each of the calibrated approach in this study, the standard deviation for every core tops associated with each method was calculated. As the relationships between oxygenation and assemblages (BFA), morphometric (MARIN), or porosity (PSA) indices for the investigated core tops are exponential (Fig 3a, b, and c), the standard deviations of the differences between the measured and the estimated [O$_2$] values were not characteristic of their method's uncertainty. For example, regarding the assemblages method, the standard deviation of the differences between the measured and the estimated [O$_2$] values for the 45 investigated core tops was 0.19 mL.L$^{-1}$. This error value is not representative of the method's uncertainty as low [O$_2$] values exhibit relatively small error (e.g. 0.04 mL.L$^{-1}$ for [O$_2$] values of about 0.20 mL.L$^{-1}$) while higher [O$_2$] values (of about 2 mL.L$^{-1}$) are likely to show bigger error (about 0.40 mL.L$^{-1}$).

To address this issue, and assess an error that is more dependent of each estimated [O$_2$] value individually, we decide to focus instead on the standard deviation of the percentage corresponding to the differences between the measured and the estimated [O$_2$] values comprised within each estimated [O$_2$] value. For the assemblages and morphometry methods, the standard

deviation based on the core tops used for their respective calibration (respectively 45 and 16 core tops) is about 23 and 24 %, respectively. As the porosity calibration is only based on 3 core tops, no standard deviation was calculated as it would not be representative.

## 4 Discussion

### 4.1 Understanding and advantages of the oxygen estimation methods

Some technical issues, specific to the morphometric analysis, are likely to induce biases during the data acquisition and processing that might result in inconsistencies in the [$O_2$] reconstruction. Broken shells, or shells that do not lay flat, might appear distorted on images, leading to inaccurate size and shape measurements. The *watershed* step used to dissociate specimens in contact with each other during the image analysis on *ImageJ*, might occasionally be incorrect, resulting in inaccurately-cut specimens and measurements. Considering these potential biases that might affect morphometric measurements, we remain highly confident with this approach given that this only happens to very few specimens (4 to 5 in average) over usually more than 300 investigated per sample.

Concerning the faunistic aspect behind the methods discussed herein, the assemblages and morphometric methods are based on the whole preserved benthic foraminiferal fauna and its variability and species succession through time according to their ecological adaptation for dissolved oxygen. As both these approaches rely on numerous species and specimens, succeeding each other (Fig. 4), according to their [$O_2$] preferences, they are likely to record and operate on a relatively large oxygen gradient, from its nearly complete depletion (as some species can survive temporarily in anoxic conditions) until it cannot be considered as a limiting factor anymore (well oxygenated conditions, usually more than about 2 to 3 mL.L$^{-1}$). However, the indices defined in the present study cannot exceed a certain threshold (e.g. the oxic assemblage cannot exceed 100 %).

Regarding the assemblages method, 10 core tops out of the 45 total, were originally investigated for their fine fraction (>61 or >63 $\mu$m; see list of cores and affiliated authors in Table 1). The results are very well integrated into the strong relationship between measured oxygen concentration and BFA index. This indicates that both fractions could successfully be used to estimate paleo-oxygenation in OMZs.

Regarding the morphometry-based approach, a link is visible between BWO and the benthic foraminiferal shape (average roundness) for each sample, from elongated species in dysoxic conditions (e.g. *Bolivina, Buliminella* species) to rounded species in oxic conditions (e.g. *Planulina* or *Cibicides* species). This relationship is however likely to reverse after a certain BWO threshold, where elongated oxic species will start to appear (e.g. *Lagena, Dentalina* species). As the later species are usually longer than the elongated dysoxic species, a size factor can thus be used to distinguish between an overall dysoxic and elongated assemblage (usually small in size) and an overall oxic and also elongated assemblage (usually big in size). This correction enables the MARIN to be gradually increase from low (less than 0.1 mL.L$^{-1}$) to high (about 2 mL.L$^{-1}$) oxygen conditions (Figs 3 and 4). It has to be noted that some rounded taxa are also adapted to extremely low oxygen values, such as *Rotaliatinopsis semiinvoluta* in the Arabian Sea (e.g. Jannink et al., 1998, ;this study), *Epistominella smithi* in the SWACM (Schmiedl et al., 1997), ENP and WNP (Tetard et al., 2017a, this study) or *Takayanagia delicata* in the ENP and WNP (Tetard et

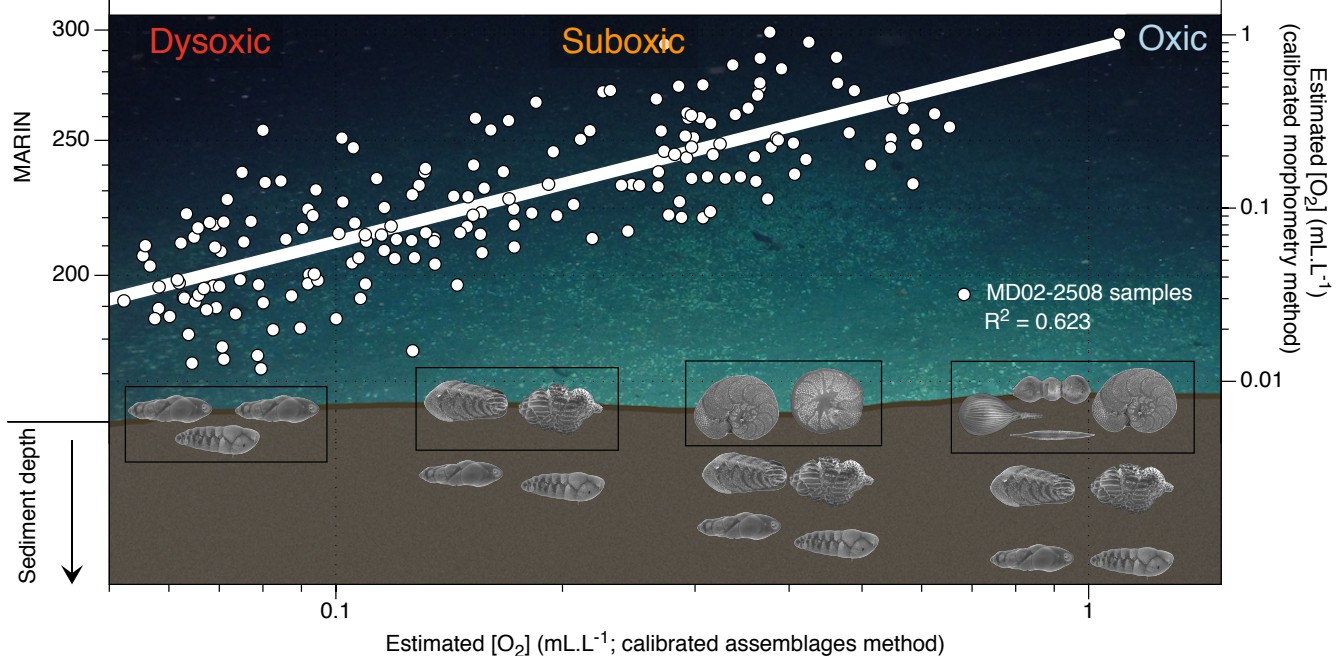

**Figure 4.** Conceptual graphic of the MARIN response of the top centimetres benthic foraminiferal assemblage to [$O_2$] variations (background image modified from Woods Hole Oceanographic Institution, courtesy of DEEP SEARCH 2018 - BOEM, USGS, NOAA). The plot shows estimated [O2] using the assemblages method vs estimated [O2] using the morphometric method (and corresponding MARIN values) for Core MD02-2508. Black boxes show species example of the top-centimetre sample gradually replaced with an increase in BWO.

al., 2017a, this study). These species will tend to indicate a more rounded assemblage and thus higher oxygen levels. However, these typical rounded species from deoxygenated regions are also relatively small by comparison with rounded species from oxic conditions such as *Planulina* or *Cibicides* species, and usually co-occur with more elongated species such as *Bolivina* and *Buliminella* species, that will overall be reflected by a relatively low MARIN and thus low oxygen concentrations. We
5  also emphasize that the presence and abundance of some shallow infaunal taxa with elongated tests (e.g. some *Uvigerina* or *Bolivina* species in the Mediterranean Sea and Atlantic Ocean) might increase under high oxygen conditions during time of enhanced organic matter fluxes (higher nutrient availability). This explains why this method should be used to investigate oxygen variability in known or supposed OMZ areas, not to assume occasional deoxygenation events.

    Concerning the porosity approach, previous studies already showed that some species react to bottom water oxygenation
10  decrease (increase) by increasing (decreasing) the pore density (Kuhnt et al., 2013) and pore surface area of their shells (Fig. 5). A closer look at the correlation between [$O_2$] and *Bolivina pacifica*'s pore density from Kuhnt et al. (2013), for example, indicates that within the estimated [$O_2$] range of Core MD02-2508 (from about 0.5 to about 1 mL.L$^{-1}$), pore density shows a limited variability. Thus, a complex response should occur over a relatively restricted oxygen gradient, where porosity probably responds to several factors at once, and therefore represents a mixed environmental signal (Glock et al., 2011). However, when

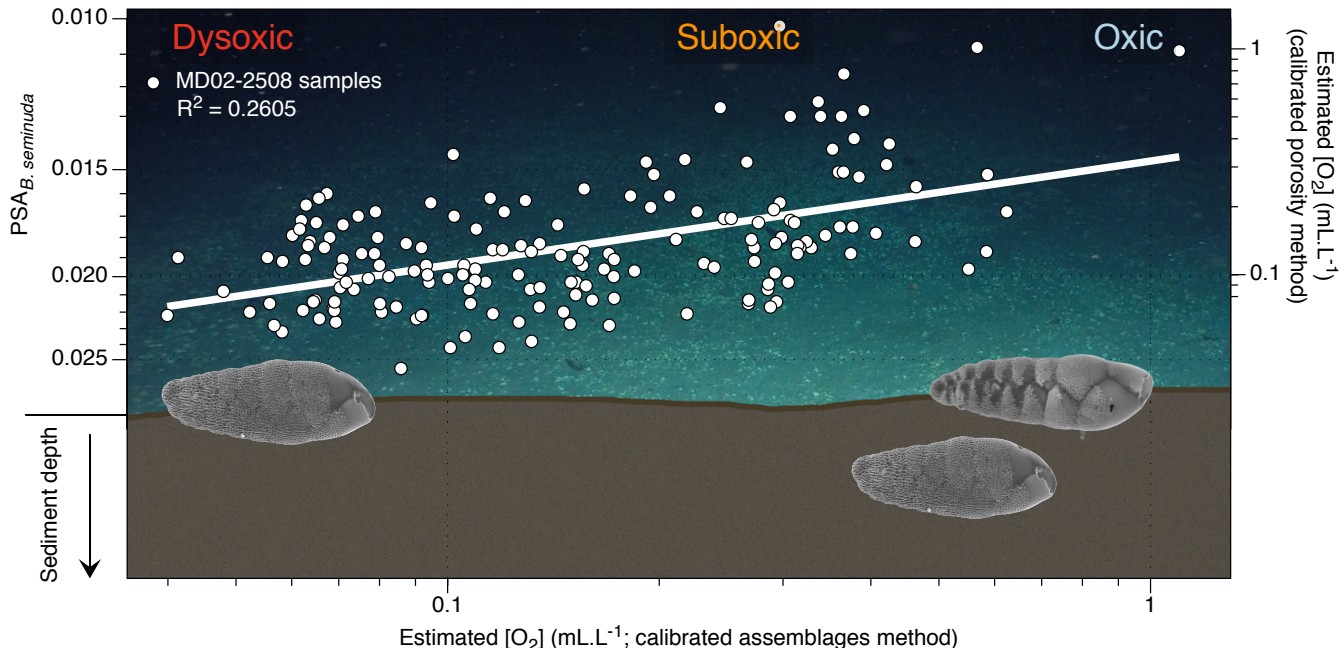

**Figure 5.** Conceptual graphic of the porosity response of benthic foraminifera (*B. seminuda* for the ENP OMZ) to $[O_2]$ variations (background image modified from Woods Hole Oceanographic Institution, courtesy of DEEP SEARCH 2018 - BOEM, USGS, NOAA). The plot shows estimated [O2] using the assemblages method vs estimated [O2] using the porosity method (and corresponding PSA values) for Core MD02-2508. Images of *B. seminuda* shows the gradually change in porosity associated with oxygen changes.

the gradient is larger, significant porosity changes might occur (Kuhnt et al., 2013) while faunal turnovers are expected to happen and the species of interest may not be found anymore. This method should thus be used for past oxygen estimation comprised within the oxygen range of predilection of the species of interest. The three methods discussed herein are then likely to be reliable in poorly oxygenated environments, but cannot ensure a consistent estimation in oxygenated environments (more

5    than about 2 to 3 mL.L$^{-1}$), or when dissolved oxygen is not the principal ecological parameter responsible for observed faunal changes.

   Regarding the recorded oxygen signal investigated with these three methods, one may question its water / sediment interface vs pore water origin. When oxic conditions prevail, the presence of potentially epibenthic species (recorded by the relative abundance of the oxic assemblage and rounded shells through the assemblages and morphometry methods, respectively) at the

10   sediment surface is considered to be representative of bottom water oxygenation. In time of oxygen depletion, these species are replaced by endobenthic (usually elongated) species migrating up to the water / sediment interface, which also become representative of bottom water conditions (Fig. 4). Conversely, during well oxygenated conditions the endobenthic species are likely to move deeper into the sediment while the epibenthic species colonise the water / sediment interface again. In this way, the interface is always occupied by an assemblage or a morphometry characteristic of a specific oxygen level. The

assemblages and morphometric methods are thus likely to be characteristic of bottom water and interface conditions (Fig. 4, black boxes). The endobenthic specimens (*B. seminuda* species) used for the porosity-based method, however, might migrate into their micro-habitat according to the redox front, and thus record either bottom water or pore water conditions depending on the level of oxygen and penetration of the front into sediment depth (Fig. 5). In addition, pore water oxygenation largely depends on surrounding bottom water oxygenation. The porosity method is thus likely to represent a mixed signal between bottom and pore water conditions.

## 4.2 Comparison of oxygenation tracers

The reliability and consistency of the three assemblages, morphometric, and porosity-based methods used for past oxygenation reconstructions in this study, and based on benthic foraminifera, were investigated on Core MD02-2508 (ENP OMZ) as this core was investigated using the three approaches at a high resolution (189 samples covering the last 80 kyr). The three methods provide similar $[O_2]$ estimations for the modern investigated core tops (Figs. 3d, 3e and 3f). First, the modern $[O_2]$ estimations based on the assemblages, morphometric, and porosity approaches (about 0.08 mL.L$^{-1}$, 0.12 mL.L$^{-1}$ and 0.16 mL.L$^{-1}$, respectively, 0.12 mL.L$^{-1}$ in average for the ENP OMZ) are very consistent with the modern mean annual dissolved $[O_2]$ value of 0.13 mL.L$^{-1}$ (Garcia et al., 2014) measured at 600 mbsl near the core site.

Throughout Core MD02-2508, these three independent approaches also exhibit similar average values (0.20 mL.L$^{-1}$, 0.14 mL.L$^{-1}$, and 0.22 mL.L$^{-1}$, respectively for the assemblages, morphometric and porosity methods). These approaches also cover similar $[O_2]$ gradients of 1.06 mL.L$^{-1}$ (from 0.04 mL.L$^{-1}$ to 1.10 mL.L$^{-1}$), 1.04 mL.L$^{-1}$ (from 0.01 mL.L$^{-1}$ to 1.05 mL.L$^{-1}$), and 1.10 mL.L$^{-1}$ (from 0.04 mL.L$^{-1}$ to 1.14 mL.L$^{-1}$), respectively. The three past $O_2]$ estimation methods thus exhibit very similar results and variations for Core MD02-2508 (Fig. 6) which are very consistent with the Northern Hemisphere climate variability record by the isotopic composition ($\delta^{18}O$) of the NGRIP ice core (Johnsen et al., 2001). As this relationship was already investigated in Tetard et al. (2017a), there will be no further discussion in this study.

As the assemblage method was calibrated based on numerous core top samples, we choose to use it as a reference for comparison with the other methods. Overall, the assemblages and morphometric approaches show similar and consistent estimated $[O_2]$ values downcore ($R^2$ = 0.62, Fig. 4) which was expected as both methods rely on the complete assemblages (census data or morphometric measurements) of each sample. The assemblages and porosity approaches exhibit a less clear but still existing relationship ($R^2$ = 0.26, Fig. 5), which can be explained by the fact that the assemblage method is based on the whole assemblage, and is likely to reflect bottom water conditions, while the porosity approach is based on porosity measurements of a single species and probably reflects mixed conditions between bottom and pore waters.

## 5 Conclusions

We conclude that the present study demonstrates the reliability of a new, fast and semi-automated morphometric analysis in OMZs, performed on benthic foraminifera for estimating past $[O_2]$, with an overall higher MARIN morphometric index (higher circularity and larger specimens) in samples characteristic of oxic conditions, while poorly oxygenated samples are associated

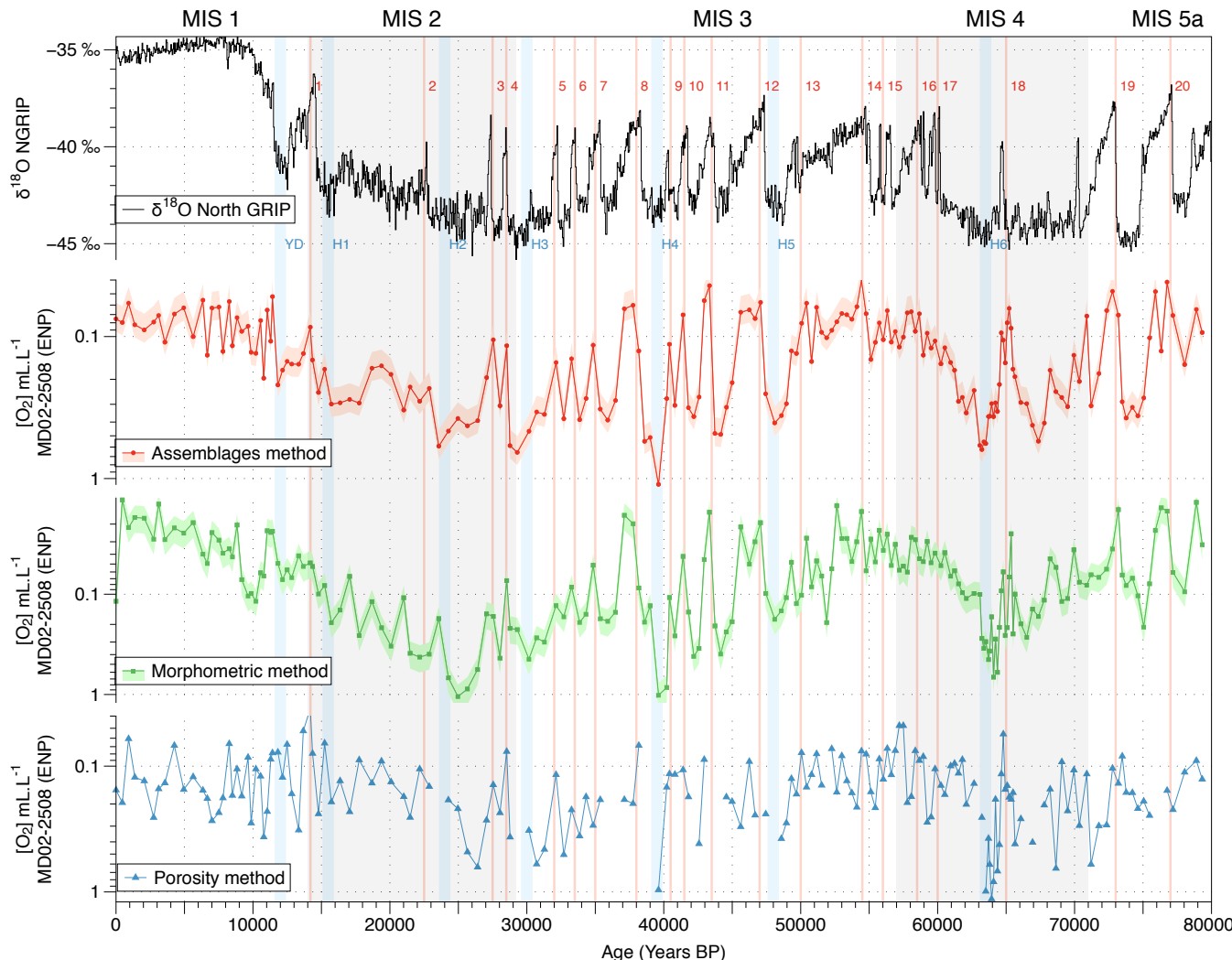

**Figure 6.** Comparison between the isotopic composition ($\delta^{18}$O record) of the NGRIP ice core from Johnsen et al. (2001) and bottom water [O$_2$] variations estimated using the calibrated assemblages (red circles), the morphometry (green squares) and the porosity (blue triangles) methods. The pink and light green shades represent the 23 and 24 % uncertainty on the assemblages and morphometry approaches, respectively.

with a lower MARIN (lower circularity and smaller size). Since only a basic taxonomical knowledge is required for this new method, its main advantages are its user-friendliness to non-specialists besides its ease and speed of image acquisition and automated processing.

5   A calibration based on numerous modern core tops (compilation of 45 cores from oxygen deficient areas recovered from all the main OMZs in the world (AS, ENP, ESP, EEP, WNP, SWACM), and along several oxygen gradients for this new approach,

as well as for the assemblages-based method from Tetard et al. (2017a) and the porosity-based method from Tetard et al. (2017b) shows similar and consistent estimations, again proving the reliability of these approaches as global past [$O_2$] tracers. However, as these methods are prone to potential specific biases, we highly encourage their combination, whenever possible, for higher reliability.

5 *Competing interests.* The authors declare that they have no conflict of interest.

*Acknowledgements.* We thank the Agence Nationale de la Recherche for its financial support to LB and MT under projects ANR-12-BS06-0007-CALHIS and JPI Blemont project ANR-15-JCLI-0003-05-PACMEDY. We thank IODP-France for its financial support to MT. This work is a contribution to the EU FP7 project MedSeA (grant agreement no. 265103). Samples from WNP were studied in the framework of the Shirshov Institute state assignment (theme No. 0128-2021-0006, EO). Many thanks to the professional captains and crews of the research-vessels *Marion-Dufresne*, *Sonne*, *Meteor*, and *Olaya*. We would also like to thank Franck Bassinot, Frans Jorissen, and Helena Filipsson for their valuable contributions to this manuscript. EO is thankful to Elena Ivanova for collecting core-top samples from WNP.

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
