# Peer review of "Toward a global calibration for quantifying past oxygenation in oxygen minimum zones using benthic Foraminifera"

_Biogeosciences, 2020_

## Author Response (AR2)

Martin Tetard
Aix Marseille Univ,
CNRS, IRD, Coll France, CEREGE,
Aix en Provence,
France.
*tetard.martin@gmail.com*

Hiroshi Kitazato
Handling Associate Editor
kitazatoh@jamstec.go.jp

April 6, 2021

Dear Mr. Editor

Thank you very much for accepting our manuscript. Please find attached the point-by-point response to every comment raised by the reviewers.

We hope that our request will receive favourable consideration,
Yours sincerely,

Martin Tetard and co-authors.

**RC1:**

**General Comments**

The authors are commended for their innovative approach to solving the "paleo-oxygen proxy problem." The index (MARIN) may be quite useful for interpretations of relatively recent changes in OMZ magnitude and extent, particularly in places where OMZ exist now. However, it is not clear how useful this method will be for studies of the deeper past (deep time) because the index "likely" reverses once a critical threshold of about 2 mL O2 / L occurs (page 12 line 3) due to inclusion of elongated oxic foraminifera. Thus, if little is known about the paleoceanographic conditions in a given region at a given time, the MARIN will not confidently identify past OMZ regions or their dynamics. Because the range of oxygen values for samples used to develop MARIN ranged only from 0.03 to 1.79 mL/L, samples from anoxic and well-aerated sites were not included as end members so the search for a ubiquitous, wide-ranging paleo-oxygen proxy remains rather elusive.

**Answer**: First of all, we thank the anonymous referee #1 for his/her constructive comments on our manuscript. Indeed, we believe that this method should be used to investigate known or suspected current and past OMZs, not to assume an unknown OMZ only based on the MARIN. A short discussion was added in the text, in the "Understanding and advantages of the oxygen estimation methods" subsection. The reversal we talk about on page 12 line 2 is the reversal of the shape measurement (roundness), not the MARIN. As the roundness is likely to reverse past a certain threshold, the size is used to correct it as OMZ species are usually small while oxygenated species could be relatively big. The paragraph page 12 was

corrected to avoid confusion. Thus, the MARIN should not be prone to the reversal observed in the roundness measurement, although we would expect a limit in the use of this index past a certain threshold (not reach at 2ml.l-1 in the new version where more core tops were added in the calibration. Samples up to 2.88 ml.L-1 were added for the assemblages calibration, however, as the relationship is exponential, no samples down to 0 ml.L-1 were added.

**Specific Comments**

**Comment**: The size fraction studied here is the >150 micron fraction (page 5 line 7). There are many publications that document how important and prevalent small sized benthic foraminifera are in low-oxygen environments (reviewed in a chapter of Sen Gupta 1999, Modern Foraminifera) Indeed, Moodley et al. (1997) specifically states that the 38-45 micron fraction is necessary to document foraminifer assemblages from low oxygen habitats.

**Answer**: 10 core tops were added to the assemblages method calibration where the investigated fraction is > 61 or > 63 μm (see table 1). The results are very well integrated into the strong relationship between measured oxygen concentration and benthic foramininferal index calculated for >150 μm size fraction. This indicates that both fractions could successfully be used to estimate paleooxygenation of a certain OMZ area. This was added to subsection 4.1.

 Regarding the morphometric method, as the fine fraction < 150 μm might exhibits broken specimens (thus not reflecting the true shape of the specimen) or juveniles (also not reflecting its final shape, as for example some macrospheric forms of bolivinids will appear round due to the size and shape of proloculus, while the adult forms will be elongated). A short discussion was added in the subsection "Morphometric analysis procedure". Regarding the 38-45 micron size fraction, we don`t have such a tiny fractions available at this stage of investigation because we believe that the majority of researchers do not use this fraction when investigating OMZ areas, however, this fact might be taken into account in future studies.

**Comment**: There is no mention of pore-water chemistry of OMZ regions. It is established that pH is lower in organic rich, low oxygen sediments compared to sediments from more aerated, lower organic settings (e.g., Sato et al 2018 Frontiers in Marine Science). This is important because the foraminifer shells of OMZ are lightly calcified and thus could be dissolved by the lower pH pore waters.

**Answer**: Indeed, there are no data on pore-water chemistry of OMZ regions studied however, however, we emphasized in the manuscript, the specimens used for calibration of the morphometric and assemblages methods were all well preserved and not broken nor dissolved (see subsection "Morphometric analysis procedure". We believe that good preservation might be related to high sedimentation rates at the site locations studied most likely due to their close position to the land where they could be affected by enhanced terrigenous material supply.

In particular, river discharge provides the semi-enclosed marginal seas are affected by intensive river discharge (Yukon, Anadyr, Kuskokwim in the Bering Sea (>157 mln tons per year) and Amur River in the Sea of Okhotsk (24 mln tons per year)) which creates favorable

conditions for fast burial of microfossils and prevent their dissolution. According to the previous studies of core SO201-2-77KL (Riethdorf et al., 2013), the average sedimentation rates vary between 9 and 16 cm/kyr reaching up 30 cm/kyr during deglaciation. Regarding the equatorial and tropical Pacific samples, this area is mostly influenced by monsoon activity which might be responsible for active terrigenous material supply to the site locations and, thus, for enhancement of sedimentation rates.

**Comment**: There is no discussion of the variability / seasonality in some OMZ zones (e.g., Arabian Sea daily to decadal variability; Banse et al. 2014 Biogeosciences). From a different perspective, it is important that the text emphasize that oxygen values for sample locales were deduced from a 2013 database, and are presented as mean annual values, while samples were largely collected in 2002. This 11 years differential could be significant given that OMZ regions and magnitudes are expanding.

**Answer**: As collected samples used for calibration of every methods presented in this manuscript originate from 1 cm thick slices which might represent several decades of sedimentation averaged, and as we didn't investigate the living fraction of foraminifera, but the whole living and dead tests, we believe that the 11 years difference between the sample collected in 2002 and their mean annual dissolved oxygen values from the 2013 database is really not significant, as most of the tests recovered in our samples would also be dead for several decades. Focusing only on the living fraction wouldn't make any sense as this method was developed to be used on the fossil fraction for past reconstruction.

**Comment**: There is no accommodation for "soft-shelled" foraminifera, the shapes/tests of which would not survive the desiccation step of the processing methods. Soft shelled foraminifera can be quite abundant in the deep sea in general (many AJ Gooday et al. publications) and OMZs (e.g. Caulle et al. 2015 Biogeosciences). Further, there is little mention of agglutinated foraminifera, which can also be quite abundant in OMZ foraminiferal assemblages (e.g. see Gooday papers).

**Answer**: As the method presented in the manuscript are meant to be easily applied to samples from the fossil records, for paleo-reconstructions, prepared following a standard procedure, the soft-shelled foraminifera were not investigated. Moreover, soft-shelled foraminifera and agglutinated tests usually do not preserved in the sediment record. Thus, they can hardly be used to help to reconstruct past oxygen concentrations. However, agglutinated and calcareous tests were not separated in this study as the whole assemblage was investigated for each sample, and some samples contain few agglutinated test (e.g. the ESP samples). This was added to the "Benthic foraminiferal assemblages study" subsection.

**Comment**: The Introduction discussed circularity but does not adequately define it, other than implying it means planispiral [note proper spelling] and trochospiral forms, noting that *Cibicides* and *Planulina* species are "completely rounded" (page 2 line 32). A completely rounded object is a sphere, yet neither *Cibicides* nor *Planulina* are spherical in form. Thus, this paragraph is not clear. Further, the term "circularity" is an odd word choice

because "elongated" serial tests are also "circular" if one views them parallel to their long axis. What is the difference between the "roundness factor" (page 3 line 7) and "circularity" (page 3 line 5)?

**Answer**: In the text, the term of circularity is used in term of test shape to assume if the test is more "elongated" of more "circular". From above in picture, the species could be spherical or planispiral, it will appear "circular" in 2D. However, the term spherical was never used. In the sense of image analysis, "circular" come from "circle" which is a 2D object. As we observe these specimens in 2D images, the term "circular" seems appropriate. The term "completely" was removed from the text, even though "completely circular" implies a circle to us, and not a sphere.

However, the term roundness refers to a specific measurement available in ImageJ and which is defined in the text. The term "planispiral" was corrected. Regarding the elongated tests that might appear circular if looked from above, in the same way that planispiral tests could appear elongated if looked from their sides, we think that the chance for these tests to stand still when dropped on the tray is very small. Some changes were applied to the introduction paragraph.

**Comment**: It is not clear why the MARIN uses the ellipse as its shape descriptor (page 6 line 31) because there are many test shapes that are not elliptical or "completely rounded" (page 2 line 32). Page 12 (lines 1-6) states that MARIN "reverses" over a certain threshold where rarer elongated species appear in oxic sediments, citing Figure 4. The figure does not show a reversal, or else the reversal needs to be explicitly labeled. Further, how was the size factor calculated (page 12 line 5)? This passage must be clarified.

**Answer**: MARIN does not use the ellipse as a shape descriptor, but the MAJOR axis measurement (automatically performed on ImageJ) does. The shape parameters retained is the roundness. The size parameters retained is the Major axis, which is not the longest axis of the specimen, but the longest axis to the fitting ellipse that have the same area as the specimen. This is commonly used to avoid error in measurement in case of some pixel that do not correspond to the specimen (error in the watershed, line due to light reflecting, for example) might biased size measurement. This is now more developed in subsection "Morphometric analysis procedure". More explanation is available in Schneider, C.A., Rasband, W.S. and Eliceiri, K.W.: NIH Image to ImageJ: 25 years of image analysis, Nature Methods, 9, 671-- 675, 2012.

We do not state that the MARIN reverse past a certain [O2] threshold, but that the roundness does, explaining why we correct it using the size to decipher between dysoxic elongated assemblages (with small tests) and oxic elongated assemblages (with a big size). This explains why no reversal is visible in Fig 4 and no reversal is occurring. The size factor was not calculated, it is automatically provided by ImageJ and correspond, as explain above to the longest axis of the ellipse fitting to the specimen. This was clarified in the text.

**Comment**: *Bolivina* is a key genus in this contribution. Most *Bolivina* tests do not have equal minor axes, meaning they can be oriented in different projections during imaging. This, along with the fact that some tests do not "lay flat" (page 11 line 4), is concerning.

**Answer**: As most Bolivina do not have equal minor axis, it means on the opposite that every specimen of each species will be oriented in the same way during imaging always laying on the same side. However, when their minor axis are equal, there is a risk that the specimens could lay on a plane associated with on minor axis, or another, but which is not of matter if the minor axis are equal, as the measurements will this be the same. The fact that some tests do not lay flat is exceptional and do not influence the average for the whole assemblage as it might only concern a few tests per sample, containing about 300 specimens.

**Comment**: It is widely known that piston cores typically do not recover the sediment-water interface (e.g. Blomqvist 1991 Mar Ecol Prog Ser; McIntyre 1971 Nature) and can "blow away" many cm of sediment. This fact is important because "the main core investigated" was collected with a piston corer (page 4 line 5).

**Answer**: The results obtained from both the "multicorer" and "piston corer" samples demonstrate consistent picture of high correlation between assemblage composition and modern O2 values, that`s why we believe that possible missing of the uppermost part of the sediments should be acceptable for this study. The main core investigated for comparing the three approaches (see figs 4, 5 and 6) is 40 m long and the results obtained by these three techniques are quite similar.

**Comment**: Please provide the type of corers used for the "several core tops" noted on page 4 line 8.

**Answer**: This sentence was emended to provide the variety of corers used in this study.

**Comment**: It is not possible to assert that "no taxonomic knowledge is required" (page 7 lines 17-18) because the person isolating the foraminifera needs to be able to distinguish between foraminifera and other items (e.g., echinoderm spines, gastropods, ostracods, planktonic foraminifera) and understand the wide variety of benthic foraminifera shapes.

**Answer**: We believe that echinoderms, gastropods and ostracods are very rare in OMZs, but indeed the person in charge should be at least able to recognize a benthic foraminifera. Regarding the wide variety of benthic foraminiferal shape, we think that it is quite limited in OMZ areas. The sentence was corrected.

**Comment**: It is not clear how species were assigned to a particular oxygen guild (i.e., dysoxic, suboxic, oxic) other than "the literature". Citations must be provided for each species assignment.

**Answer**: An appendix was added to provide citations for each species assignment (Table S2).

**Comment**: The figure captions are too general and do not explain what is illustrated / plotted. This is especially true in Figures 4 and 5, which plot Estimated [$O_2$] versus Estimated [$O_2$]. Further, the source of the plotted data is not explained, the black boxes are not explained, and the significance of the foraminiferal images is not explained.

**Answer**: Captions were emended. Fig. 4 plots estimated [O2] using the assemblages methods vs estimated [O2] using the morphometric method, to compare both approaches. Fig. 5 plots [O2] estimation from assemblages and porosity approaches for comparison. The source of the plotted data is the current study. Black boxes and foraminifera images show a typical assemblages from Core MD02-2508 during Oxic, suboxic, and dysoxic conditions. This was added in the respective captions.

**Technical Errors**

**Comment**: Verb tense should be past tense instead of present tense when describing what occurred during analyses (e.g. page 4 line 5 should read "Residues were then rinsed…").

**Answer**: The whole preparation procedure was corrected to past tense.

**Comment**: Many passages assert points but are not supported by literature citations (e.g. phrase ending in "OMZ" on page 3 line 5; page 11 lines 13 and 14).

**Answer**: Citations were added.

**Comment**: The term "preference" and "prefer" is used many times throughout the contribution (e.g. page 2 lines 15, 17, 28) in the context of where a foraminifera species lives. Foraminifera are not capable of conscious thought and thus cannot prefer one environment over another.

**Answer**: The term "preference" was used as it was originally from a title of one study used in this study: Corliss, B. H.: Morphology and microhabitat preferences of benthic Foraminifera from the northwest Atlantic Ocean, Mar. Micropaleontol., 17, 195--236, 1991.

It was nevertheless corrected for "adaptation" in the text.

**Comment**: Page 2, line 12: fossil benthic foraminifera were not reported in Bernhard and Riemers 1991.

**Answer**: This was corrected for "Benthic Foraminifera".

**Comment**: Do not italicize "and" (page 2 lines 32 and 34).

**Answer**: This was corrected.

**Comment**: The term "perfect" is generally not used in scientific literature (page 2 line 32).

**Answer**: The term was modified

**Comment**: "Burry" should be "bury" (page 3 line 1).

**Answer**: This was corrected

**Comment**: "This" needs to be defined on page 3 line 9.

**Answer**: This was corrected.

**Comment**: Line 15 of page 3 should read "…of each of the three existing BWO estimation methods, based on either assemblage composition, porosity, or morphometry."

**Answer**: This was corrected

**Comment**: Page 4 line 21 should read "…counts is available in their supplementary…".

**Answer**: This was corrected

**Comment**: Italicize all ship names (page 4 line 22).

**Answer**: All ship names were italicized.

**Comment**: Page 5 line 15 should read "…very abundant throughout Core…"

**Answer**: This was corrected.

**Comment**: Page 5 line 16 should read "…and its ecology is well documented in the literature…".

**Answer**: This was corrected.

**Comment**: The assertion that the morphometric analysis is "taxon-free" (page 6 line 10) is nonsensical because "taxon free" indicates that zero taxa are included in the analysis, which is not the case. Perhaps the authors intend to state "taxon independent"?

**Answer**: This was corrected for taxon-independent.

**Comment**: It is alarming that the original image is deleted as part of the protocol (page 7 lines 1-2).

**Answer**: Before the submission of the manuscript, this step was made optional and the original image are not deleted anymore (although the user can choose to make it happen again). This was originally design for keep as many storage space as possible by deleting the original images which can be quite heavy, as the resized images are saved.

**Comment**: Why do the equations use "Pi" instead of the universally accepted symbol for pi?

**Answer**: This was corrected by adding the symbol.

**Comment**: Equations typically are not presented in italics font, although perhaps *Biogeosciences* allows or demands this.

**Answer**: The italics font is automatically generated by the equation function in Latex.

**Comment**: Provide detailed identification information for sample shown in Figure 2.

**Answer**: The information for this sample was added in the legend of Figure 2. The illustrated sample originates from Core MD02-2508, sample 240-241 cm

**Comment**: The Results section is confusing because it presents detailed passages only to end the paragraph indicating that that approach was abandoned (e.g., Section 3.1).

**Answer**: We wanted to give an history of the development of the approach, for example in the first paragraph of subsection 3.1, the principle of the method was explained and ended with the fact that it was needed a calibration which is now presented in the current study. It was emended for clearer explanation.

**Comment**: How is oxygen measured with a CTD (C = conductivity / salinity; T = temperature; D = depth / pressure; e.g. page 8 line 12)?

**Answer**: It was measured using a CTDO, it was corrected in the text.

**Comment**: Assign approximate values to "to very few specimens" (page 11 line 8).

**Answer**: This was corrected. After investigation of several processed images, "very few specimens" is usually 4 to 5 specimens over about 300. It was added in the text.

**Comment**: Inclusion of "whole" in "…on the whole benthic foraminiferal fauna…" (page 11 line 10) is inappropriate because soft shelled foraminifera are not considered in this morphometry method.

**Answer**: This was corrected for "the whole preserved benthic foraminiferal fauna".

**Comment**: The parenthetic on page 12 line 10 defines all possible values (through infinity) because it includes values under 0.1 through those exceeding 1 (no upper limit).

**Answer**: This was corrected.

**Comment**: Page 12 line 11 should read "…response should occur over a relatively restricted…"

**Answer**: This was corrected.

**Comment**: Page 12 line 16 should read "…in oxygenated environments…" (add "s" to environment).

**Answer**: This was corrected.

**Comment**: Page 13 line 6 should read "Throughout Core MD02…"

**Answer**: This was corrected.

**Comment**: Where is "downcore" in Figure 4, per comment on page 13 line 11?

**Answer**: This was corrected in the legend.

**Comment**: Some statements in the Conclusions need qualification: "…no taxonomical knowledge is required" (page 14 line 5) was discussed elsewhere in this review; "…oxygen deficient areas recovered from all over the world" (page 14 lines 8-9) is inaccurate as there are no samples from many areas, such as the Atlantic and Indian Oceans (e.g. OMZ off Namibia).

**Answer**: The taxonomical requirement was detailed. The conclusion was also emended regarding the fact that we used numerous (45) modern samples recovered from all the main OMZs in the world (AS, ENP, ESP, EEP, WNP, SWACM).

**RC2:**

**Comment:** The authors explore the relationships between various morphological characteristics of benthic foraminiferal tests (porosity, size distribution, circularity) and the ambient bottom-water oxygen concentration. In this context, the authors established calibration data sets for OMZ samples mainly from the Pacific Ocean and successfully applied the new method to a paleo-record from the eastern Pacific Ocean. Oxygen plays a crucial role in shaping the ecosystem diversity and species composition of marine environments and oxygen changes are tightly linked to climate variability and marine biogeochemical cycles. Recent studies have documented an extension of dysoxic zones in modern ocean environments due to the effects of global warming. In order to assess the potential future impacts of ocean deoxygenation, it is very important to better understand how ocean environments and marine ecosystems responded to climate-related oxygen changes in the past. The present manuscript addresses this important and up-to-date topic and since oxygen is tightly linked to nitrogen and carbon cycling, this issue is very well suited for the journal "Biogeosciences". Although the easy-to-apply morphological approach is very promising and thus highlights the significance of this study, the manuscript will profit from minor revision concerning the evaluation and presentation of the results.

General comments:

1) **Comment:** The authors demonstrate the successful application of the different morphological approaches particularly to the Pacific OMZ and future application to other OMZ appears promising. Nevertheless, at this point, a "global" calibration (as promised in the title) should be based on a much higher number from all major OMZs. Therefore, a more specific title would provide a more honest reflection of the content of the manuscript.

**Answer:** First, we thank the anonymous referee #2 for his/her constructive comments on our manuscript. The calibration of the assemblages method is now based on 45 samples recovered from the main OMZs worldwide: the ESTNP, ETNP, EEP, ESP, SWACM, and AS, only the BB is missing, (Paulmier and Ruiz-Pino, 2009). 17 core tops from the EEP were added to the calibration as well as 5 core tops from the Atlantic. This method was applied to ENP records in this study, to ENP, AS and WNP records in a manuscript that is in preparation, and was applied to WNP records in a manuscript submitted to Frontiers in Earth Sciences (Ovsepyan et al., Evidence for different intermediate-and deep-water oxygenation history in the subarctic North Pacific throughout the last deglaciation). As this method is based on the relative abundance of assemblages defined in OMZs, we are quite confident it can be applied everywhere. Regarding the morphometric approach it is indeed based on less samples recovered from less OMZs, as not all samples used for the calibration of the assemblages approach could be imaged for the morphometric approach. This is why we didn't want to name our manuscript "A global calibration…" but "Toward a global calibration…" to show the effort made in this way.

2) **Comment:** Different OMZs of the global ocean have existed over various time intervals, leading also to the evolution and establishment of new, and partly endemic, taxa. In this context, some taxa have developed and adapted to extremely low oxygen values, which do not share the expected typical morphological features usually observed in low-oxygen faunas. Just to mention some examples: In the Arabian Sea, the rounded Rotaliatinopsis semiinvoluta can occur at considerable numbers at very

low oxygen concentrations (e.g., Jannink et al., 1998; DSR I 45, 1483-1513). Similarly, the round Epistominella smithi apears co-occurs with elongated taxa in the OMZ off SW-Africa (e.g., Schmiedl et al., 1997; MarMic 32, 249-287). This potential bias should be addressed in the discussion chapter.

**Answer:** Indeed, this is also what is occurring in the ENP and WNP OMZs with the species Takayanagia delicata which is a small and rounded species. In the ESP, Nonionella species are usually rounded and big, but in these areas, the presence of Bolivina (e.g. pacifica / seminude), Eubulimina (e.g. exilis in the AS) also impact the overall morphology of the assemblages. The species you mentioned in your comment are also relatively small and then, will make the MARIN lower and indicator of less oxic conditions. A short discussion was added about this, in the subsection "Understanding and advantages of the oxygen estimation methods".

3) **Comment:** The number of elongated tests can also significantly increase in well-oxygenated environments which experience enhanced organic matter fluxes. There are many examples from shelf and slope environments (e.g. in the Mediterranean Sea and Atlantic Ocean), where the proportion of elongated shallow infaunal taxa (e.g. certain uvigerinids, buliminids and bolivinids) increase in abundance under constantly high oxygen levels. Although this potential bias has been shortly addressed in the discussion chapter it would deserve a bit more attention since it may also play a role at the upper and lower margins of OMZs.

**Answer:** A short discussion was added in subsection 4.1 to address this issue. More data would be needed to fully address this hypothesis, however, we believe this is why this method should be used to investigate oxygen variability in known OMZ area, not to assume if an area was deoxygenated or not.

4) **Comment:** The significance and applicability of your calibration data set for the new morphometric index (MARIN) is hampered by the under-representation of calibration points above oxygen concentrations of 0.3 mL L-1 (Values above ~0.3 mL L-1 are represented only by one data point). Please address this issue more elaborately in the revised manuscript. In addition, you should provide the standard deviations and associated errors of your estimates for better assessment of the uncertainty of the different transfer functions.

**Answer:** Regarding this issue, 3 more core tops investigated by E. Ovsepyan were imaged, and analyzed by our system for morphometry and were added to the calibration of the MARIN. [O2] values for these core tops range from 0.83 to 1.79 ml.L-1, significantly improving the oxygen range used for calibration of this method. Standard Deviation were calculated for the difference between expected O2 and estimated O2. However, as the function is exponential and ranging from very low (about 0.03 ml.l-1) to relatively high values (about 2 ml.l-1), the standard deviation is largely biased toward the high values which are now numerous. The standard deviation based on the absolute difference between the expected and estimated values, which is about 0.19 ml.l-1, is not very informative as for the very low values, as the error is likely to be smaller for low values (e.g. 0.10 ml.l-1 +- 0.02 ml.l-1), while it will be higher for the more oxygenated values (e.g. 2ml.l-1 +- 0.30 ml.l-1). To address this issue, we decided to focus on the standard deviation for the relative difference (in %) between the expected and estimated values which will now give a better approximation of the error for each estimated value (+- 24% for both the morphometry and assemblages

approaches). A subsection "Estimating the measurement uncertainties of the calibrated methods » was added on page 12 to discuss about the uncertainty.

Specific comments:

Abstract

- **Comment:** Lines 2-5: The first three sentences have an introductory character and may be deleted from the abstract.

**Answer:** The second and third sentences were moved to the first paragraph of the introduction. The first sentence was kept to quickly define an OMZ.

- **Comment:** Lines 13-14: It would be useful to add information on the standard deviations and thus uncertainty of the transfer functions.

**Answer:** A whole subsection is now dedicated to this issue (page 12).

1 Introduction

- **Comment:** Page 2, line 2: provide reference for the defined oxygen value of 0.5 mL L-1

**Answer:** Reference was added.

- **Comment:** Page 2, lines 25-30: you should add one or two sentences that despite the common preference of specific morphologies in low-oxygen environments there are various exceptions (see general comment above).

**Answer:** See comment above

- **Comment:** In many oxic environments, the proportion of shallow-infaunal taxa can be quite high relative to the amount of available food. A good example is the present Mediterranean deep sea, where oxygen concentrations are high but where a W-E food gradient is reflected by the proportion of shallow infaunal taxa (e.g., De Rijk et al. 1999, JFR 29, 93-103; De Rijk et al. 2000, MarMic 40, 151-166).

**Answer:** See comment above, we now discuss the presence and abundance of shallow infaunal taxa under high oxygen conditions, and high availability of -food.

2 Material and methods

- **Comment:** General remark: Concentration on the size fraction >150 μm appears useful, since this fraction is widely used in foraminiferal studies. Nevertheless, you should add,

that in future studies, the calibration should be also tested on the smaller size fraction (63-150µm) since low-oxygen environments often contain a high number of small-sized taxa and individuals.

**Answer:** Regarding the morphometric approach, we decided to focus the coarse fraction (> 150 µm) as it is the size fraction commonly used in benthic foraminiferal studies and in order to propose a calibration adapted to most of the studies. Moreover as the fine fraction (63 – 150 µm) is usually composed of fragmented / broken tests (which do not reflect the true morphology of the organisms when they were living) or juvenile specimens (which do not always reflect the true morphology of the adult forms (e.g. for *Bolivina* species, the proloculus is usually very rounded while the adult specimen can be very elongated). This was added in the manuscript page 7 in the "Morphometric analysis procedure" subsection. Regarding the assemblages approach, we mainly focused on the > 150 µm fraction, but also add 9 core tops where the > 63 µm fraction was investigated. This was added in the text in the "Benthic foraminiferal assemblages study" subsection, page 6 and in Table 1.

- **Comment:** Page 4, line 21: "...available in their..."

**Answer:** This was corrected.

- **Comment:** Page 5, line 3: "...disaggregate..."

**Answer:** This was corrected.

- **Comment:** Page 5, line 11: Provide information if specimens have been picked from splits or from the entire residue?

**Answer:** More information was added at the end of this paragraph.

- **Comment:** Page 6, lines 2-3: provide a concise summary of the method (no details but just on the general method)

**Answer:** This paragraph was re-written to provide a better explanation of the method.

3 Results

- **Comment:** Pages 8-10: as mentioned already above, please add the standard deviation to each determination coefficient in the text and in Fig. 3, and errors to estimated values in Table 2 for better assessment of the uncertainties of your transfer functions.

**Answer:** See answer above.

4 Discussion

- **Comment:** Figures 4 and 5: The graphical design of these figures should be revised. I recommend to simplify the figure by deleting the background coloration and enhancing the contrast of displayed foraminifera.

**Answer:** Background coloration of the sediment was removed in order to simplify the figure. The contrasts of the foraminifera was already enhanced as these foraminifera images were originally prepared for plates, however, as the sediment background is now darker, foraminifera are more visible.

- **Comment:** Page 12, lines 1-6: Please discuss the potential bias of food flux changes in highly oxygenated environments. Elevated numbers of elongated infaunal taxa (e.g. certain Uvigerina species etc.) may occur under similar oxygen concentrations but different food availability. This also illustrates the limitation of your approach in suboxic to oxic environments. You should clearly emphasize this in the revised version.

**Answer:** As mentioned above, this issue was addressed in the text.

- **Comment:** Page 13: You should create a new figure presenting the down-core records and oxygen reconstructions of core MD02-2508. This would further illustrate the general applicability of your new method.

**Answer:** A new figure was added to illustrate the comparison between the three methods. However, no further paleoceanographic interpretation were discussed as the downcore oxygen record was already interpreted in Tetard et al., 2017.

- **Comment:** Page 13, line 13: "...by the fact that..."

**Answer:** This was corrected.